**Data Availability Statement:** All relevant data are within the manuscript and its supporting information files.

**Funding:** Rasheed Gbadegesin is supported by National Institutes of Health/NIDDK grants

# Changing epidemiology of nephrotic syndrome in Nigerian children: A cross-sectional study

Christopher I. Esezobor[1,2]*, Adaobi U. Solarin[3,4], Rasheed Gbadegesin[5,6]

**1** Department of Paediatrics, Faculty of Clinical Sciences, College of Medicine University of Lagos, Idi-Araba, Lagos, Nigeria, **2** Department of Paediatrics, Lagos University Teaching Hospital, Idi-Araba, Lagos, Nigeria, **3** Department of Paediatrics, Faculty of Clinical Sciences, Lagos State University College of Medicine, Ikeja, Lagos, Nigeria, **4** Department of Paediatrics, Lagos State University Teaching Hospital, Ikeja, Lagos, Nigeria, **5** Division of Nephrology, Departments of Pediatrics and Medicine, Duke University Medical Center, Durham, North Carolina, United States of America, **6** Duke Molecular Physiology Institute, Duke University Medical Center, Durham, North Carolina, United States of America

* esezobor@gmail.com

## Abstract

### Background

Recent reports from small studies in West Africa suggest that Black children may have high rate of steroid sensitivity nephrotic syndrome (SSNS) contrary to long held knowledge. Herein, we determined the proportion of children with idiopathic nephrotic syndrome (INS) who achieved complete remission with steroid therapy and identified factors associated with complete remission.

### Methods

We reviewed the medical records of 241 children with INS in two centres in Lagos from 2010 to 2019. We extracted demographic data, clinical features, laboratory values at the time of diagnosis, and receipt and response to steroids and other immunosuppressants.

### Results

The median (interquartile range) age at diagnosis of INS was 5.1 (3.0–8.7) years and boys were 60.2% of the study population. Children with SSNS made up 85.9% (n = 207) of the study cohort. Among those aged 0–5 years, 92.6%were SSNS compared with 69.2% in those aged 11–17 years at the time of diagnosis. In addition, the proportion of children with SSNS increased from 73.8% between year 2010 and 2012 to 88.4% afterwards. Also, children with SSNS had lower serum creatinine (0.44 vs 0.70; p<0.001) and higher estimated glomerular filtration rate (101 vs 74.3 ml/min/1.73 m$^2$; p = 0.008) at the time of diagnosis than those with steroid resistant nephrotic syndrome (SRNS).

5R01DK098135, 5R01DK094987 and Doris Duke Charitable Foundation Clinical Scientist Development Award, and Doris Duke Clinical Research Mentorship award. The funders had no role in study design, data collection and analysis, decision to publish, or preparation of the manuscript.

**Competing interests:** The authors have declared that no competing interests exist.

## Conclusion

Among Black children in Lagos, the proportion with SSNS is comparable to proportions described in children of Asian and European descent. Furthermore, children with SSNS had lower serum creatinine and higher glomerular filtration rate than those with SRNS.

## Introduction

Nephrotic syndrome is a common kidney disease affecting approximately 2–16 per 100,000 children each year [1, 2]. It is characterised by generalised oedema, massive proteinuria and hypoalbuminaemia. Although, it has been described in all races, South Asian children are thought to have the highest incidence [1, 2]. About 80% of children with nephrotic syndrome respond to a 4 to 6-week course of prednisolone with complete resolution of proteinuria [3]. In the landmark International Study of Kidney Disease in Children (ISKDC), about 90% of the children with steroid sensitive nephrotic syndrome (SSNS) had minimal change disease, hence, the current approach to determine response to steroids before considering kidney biopsy in children with idiopathic nephrotic syndrome [3]. About 20% of children (10–20%) with nephrotic syndrome fail to respond to a standard course of prednisolone and are termed as having steroid resistant nephrotic syndrome (SRNS). Most children with SRNS have focal segmental glomerulosclerosis (FSGS) and about 10–30% are reported to have a single gene disorder as the cause of the nephrotic syndrome [4]. These children require additional second line agents such as calcineurin inhibitor (CNI), and monoclonal antibodies to achieve complete or partial remission of the nephrotic syndrome and halt disease progression [5, 6]. In the absence of remission following treatment with immunosuppressants, most children with SRNS/FSGS develop progressive disease and will reach end stage kidney disease within 5 to 10 years of diagnosis [7, 8].

Studies in multiracial countries such as the United States of America and South Africa indicate that children of Sub-Saharan African (SSA) ancestry have steroid resistance rates 2–4 times higher than rates in children of Asian and European descent [9–12]. These observations influence the management of Black children with nephrotic syndrome including discussions about its clinical course and long-term prognosis with the families [13]. Although studies in the 1970s and 1980 documented 40–70% steroid resistance rates in children in SSA [14–16], recent studies indicate high and increasing rates of steroid sensitive NS [17–19]. However, many of these recent studies were limited by small sample size, short study period, non-standardised definitions of therapy response and being single-centre studies. In this report, we described the epidemiology of INS in children attending two large academic hospitals in Lagos, Nigeria. Specifically, we aimed to determine the frequency of steroid sensitive nephrotic syndrome and its association with age at onset of the disease, sex and other clinical and laboratory features.

## Methods

The study described in this report was conducted in two large public-funded academic hospitals in Lagos, south west Nigeria. The two hospitals, Lagos University Teaching Hospital (LUTH) and Lagos State University Teaching Hospital (LASUTH) serve as the referral hospitals for Paediatric Nephrology Services in Lagos State which has an estimated urban population of 14.4 million people [20]. Although, majorly populated by people of the Yoruba ethnic

group, Lagos state, being the economic capital of Nigeria, is cosmopolitan and it is home to all major and most minor ethnic groups in Nigeria [21].

We reviewed the medical records of all children who developed nephrotic syndrome between January 2010 and December 2019 in LUTH, and between January 2015 to December 2018 in LASUTH. These start dates were chosen based on the time two of the investigators joined the services of these hospitals. As per hospital policy, children aged 0–18 years and 0–13 years are managed by paediatric services in LUTH and LASUTH, respectively. About 67% of the children in the study were also part of an ongoing genetic study of nephrotic syndrome in children. We excluded children with a secondary cause of nephrotic syndrome. We extracted from the medical records data about the age at onset of the disease, presence of haematuria, blood pressure, and result of serum albumin, cholesterol, and creatinine at the time of diagnosis of nephrotic syndrome. Other information of interest extracted included the ethnic group of the family, response to steroid, other immunosuppressants, and result of kidney biopsy.

### Definitions and management of nephrotic syndrome in the two centres

Nephrotic syndrome was defined by the presence of generalised oedema, massive proteinuria ($\geq$3+ dipstick proteinuria, >1000 mg/m$^2$/day of proteinuria on 24-hour urine or urine protein-creatinine ratio >2 g/g) and hypoalbuminaemia (serum albumin <25 g/L). Children with new-onset INS received oral prednisolone at a single daily dose of 60 mg/m2/day (to a maximum of 60 mg/day) for 4–6 weeks. If remission was achieved, at the end of 4–6 weeks, prednisolone dose was reduced to 40 mg/m$^2$ (to a maximum dose of 40 mg/day) for another 4–6 weeks. Thereafter, it was tapered over 2–3 months. Since 2013, we defined SRNS as failure to achieve remission following 8 weeks of oral prednisolone at a dose of 60 mg/m$^2$/day (maximum of 60 mg) in line with the Kidney Disease: Improving Global Outcomes glomerulonephritis (KDIGO GN) guideline published in 2012 [22]; before 2013 we used a cut off of 6 weeks to define steroid resistance. Frequent relapse and steroid dependence FRSD) were also defined according to the KDIGO GN guidelines. Steroid resistance was the major indication for kidney biopsy; in LASUTH, children with FRSD were also considered for kidney biopsy before receipt of second line drugs. In both centres, calcineurin inhibitors was the drug of choice for SRNS. Oral levamisole or oral or intravenous cyclophosphamide, depending on the choice of parents, were used as steroid-sparing agents for children with FRSD. Glomerular filtration rate was estimated using the Schwartz formula [23] [(*height*$^*$*k*)/*serum creatinine*] with 0.413 for *k*.

### Ethics approval

The study received approval from LUTH Health Research Ethics Committee and LASUTH Health Research Ethics Committee. For children who participated in the genetic study (still ongoing), written informed consent was obtained from the caregivers and assent from children older than 12 years as part of the requirements of the genetic study. For those who did not take part in the genetic study, the requirement for informed consent was waived because only previously collected information in the medical records were accessed; no contact was made with them as part of this study. All collected data was anonymised to maintain confidentiality. The study was conducted in accordance with the principles espoused in the Helsinki Declaration for the protection of human subjects in research.

### Data analysis

We analysed the data using IBM's Statistical Package for Social Sciences, SPSS, version 25 (Armonk, New York, USA). We used proportions to summarise categorical data and median

(with interquartile range) for continuous data because of their skewed distribution. We approximated age at diagnosis to the nearest year before categorising into age groups 0–5, 6–10 and 11–17 years. We compared children with SSNS with those with SRNS using Chi Square test and Mann Whitney U test for categorical and skewed continuous data, respectively. For analysis involving blood pressure, haematuria, serum albumin, cholesterol, and creatinine, we only included children who were diagnosed with nephrotic syndrome in the two hospitals or whose referral letter contained these variables of interest. P value <0.05 in two tails was considered as statistically significant in all analysis.

## Results

### Demographics

We excluded 28 children because of secondary causes of nephrotic syndrome (n = 15), unknown response to steroids because of lost to follow up (n = 11) and missing medical records (n = 2), leaving 241 for analysis (Fig 1). Over half of the children (n = 136; 56.4%) developed nephrotic syndrome before the age of 6 years; two children developed nephrotic syndrome before 1 year. Nephrotic syndrome was more common in males with a male-female ratio of 1.4 in those younger than 6 years and a ratio of 2.5 in those aged between 11 and 17 years (Fig 2).

### Response to steroid, kidney histology and use of other immunosuppressants

Of the 241 children, 207 (85.9%) had SSNS including 31 (15%) children with FRSD (Fig 3). Twenty-three children underwent kidney biopsy predominantly for SRNS (n = 18). The most common histology was focal segmental glomerulosclerosis (n = 19); others were mesangioproliferative glomerulonephritis (n = 2), and one each of membranous nephropathy and minimal change disease. Forty-eight children (19.9%) received other medications for either SRNS or steroid sparing. Calcineurin inhibitors, especially cyclosporine was the most common non-steroid immunosuppressants children received (n = 31); 13 (5.4%), 12 (5.0%) and 6 (2.5%) children also received levamisole, cyclophosphamide, and mycophenolate mofetil, respectively, with some children receiving more than one medication. Only 2 children received rituximab (one for SRNS, another for steroid dependence).

### Characteristics of children with steroid sensitive and steroid resistant nephrotic syndrome

The proportion of children with SSNS declined significantly from 92.6% in those less than 6 years to 69.2% in those older than 10 years; p = 0.001 (Fig 3). Furthermore, the children with SSNS were about half the age of those with SRNS at the onset of nephrotic syndrome (4.8 v 9.3 years; p value<0.001), Table 1. There was no sex difference in response to steroid. Steroid sensitive nephrotic syndrome was more common in children of the Yoruba ethnic group (92.3%) than in those belonging to the Ibo (80.2%) or 'Other' tribe (74.3%), p = 0.010. Similarly, there was a higher frequency of SSNS in children with onset of nephrotic syndrome after year 2012 (88.4% vs 73.8%; p = 0.013). In addition, there was a trend for higher steroid response in children managed in one of the hospitals (92.3% vs 83.5%; p = 0.082) which disappeared when analysis was limited to only children who developed nephrotic syndrome after 2014 (92.3 vs 89.1%; p = 0.495).

Including only the 208 children with available serum creatinine value at diagnosis of nephrotic syndrome, those with SSNS had lower serum creatinine (0.44 vs 0.70 mg/dL), higher

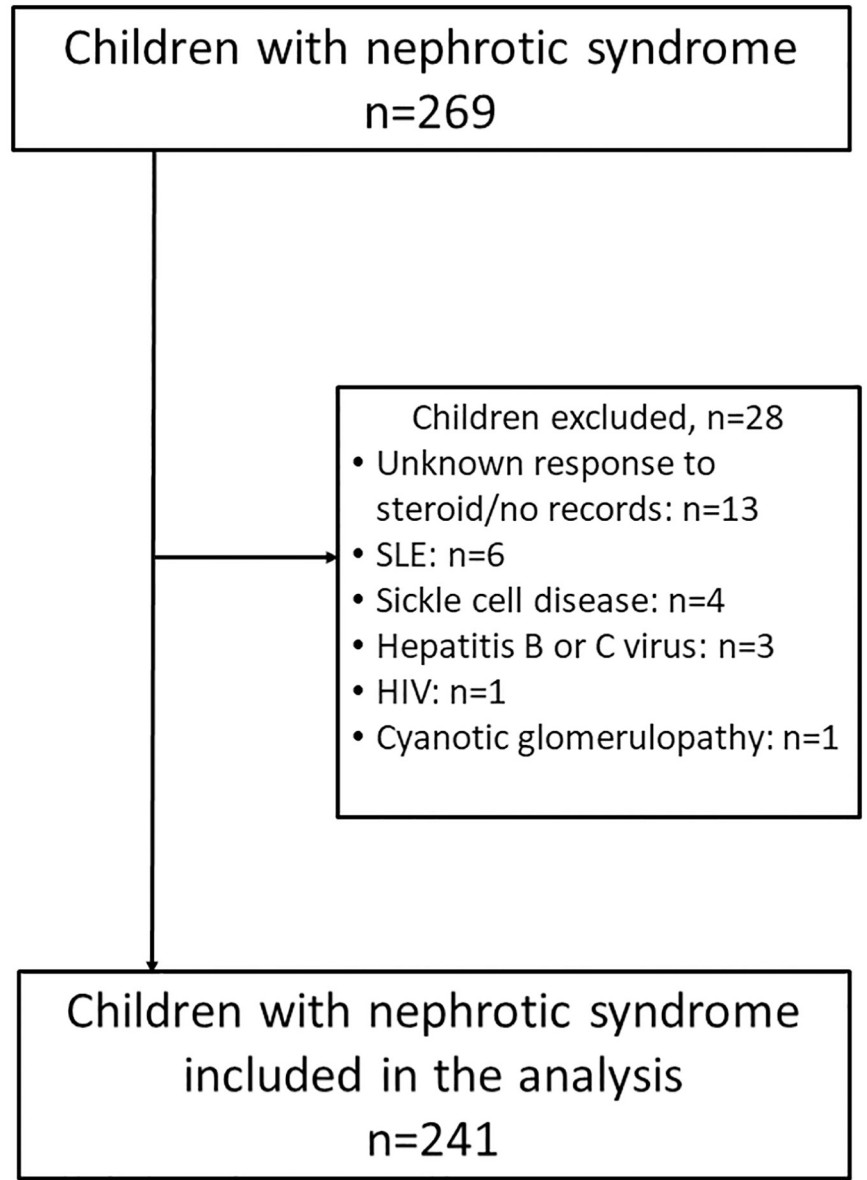

**Fig 1. Flow diagram of study participants.**

eGFR (101.0 vs 74.3 ml/min/1.73) and more hyperfiltration (98.3% vs 1.7%) than those with SRNS. However, the presence of elevated blood pressure, haematuria, or serum levels of albumin and cholesterol at the time of the diagnosis was not different in children with SSNS or SRNS.

## Discussion

Historically, Black children are known to have high rates of SRNS and FSGS compared with children of Asian and European descent. As a result, they are more likely to undergo kidney biopsy, and receive more toxic medications such as calcineurin inhibitors to induce remission when they develop nephrotic syndrome. However, recent small reports, mostly from West Africa suggest increasing rates of SSNS among children with nephrotic syndrome [17, 24, 25].

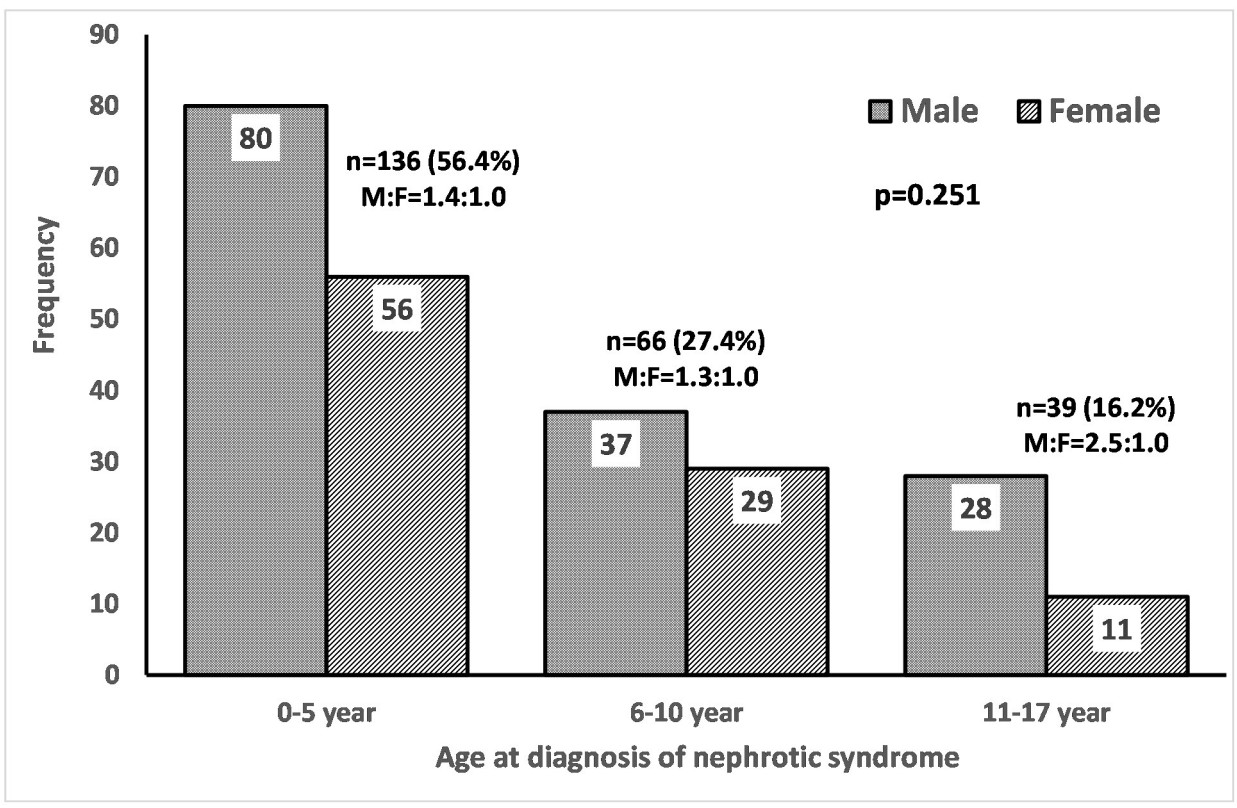

**Fig 2. Age and sex distribution of idiopathic nephrotic syndrome in children.**

In this report, we took advantage of a well characterised cohort of children with INS in two centres in Nigeria to determine the rate of SSNS in the last 10 years. In this relatively large study in southwest Nigeria, we documented that about 85% of all children and about 90% of those aged less than 6 years at the onset of nephrotic syndrome achieved complete remission following treatment with only prednisolone. This high response in our study persisted until the beginning of adolescence when it declined below 70%. The high rate of SSNS was seen across both genders, in both centres, and had increased from about 70% to about 90% after 2012.

To our knowledge, this is one of the largest studies demonstrating that Black children in SSA may also have high rate of SSNS comparable to children with Asian and European ancestry. High SSNS rate has been recently reported among Black children, albeit in relatively small studies. For example, Aloni et al. [18] reported that 72.5% of 61 children with NS seen between 1983 and 2008 at a tertiary hospital in Democratic Republic of Congo had SSNS. Another study in Côte d'Ivoire also documented that 84% of 108 children with nephrotic syndrome had SSNS [17]. Our study, together with these studies, indicates that high steroid sensitivity may no longer be uncommon among Black children in SSA and that when they develop nephrotic syndrome they should be treated like children with Asian and European ancestry. In addition, the high rate of SSNS should also guide discussion with the families of Black children with nephrotic syndrome in terms of expectation of a response to a standard course of steroid. However, our observation contrasts sharply with the long-held understanding from many studies in multi-racial communities. For example, in a recent report from Johannesburg, South Africa, only 54% of Black children with nephrotic syndrome had SSNS compared with

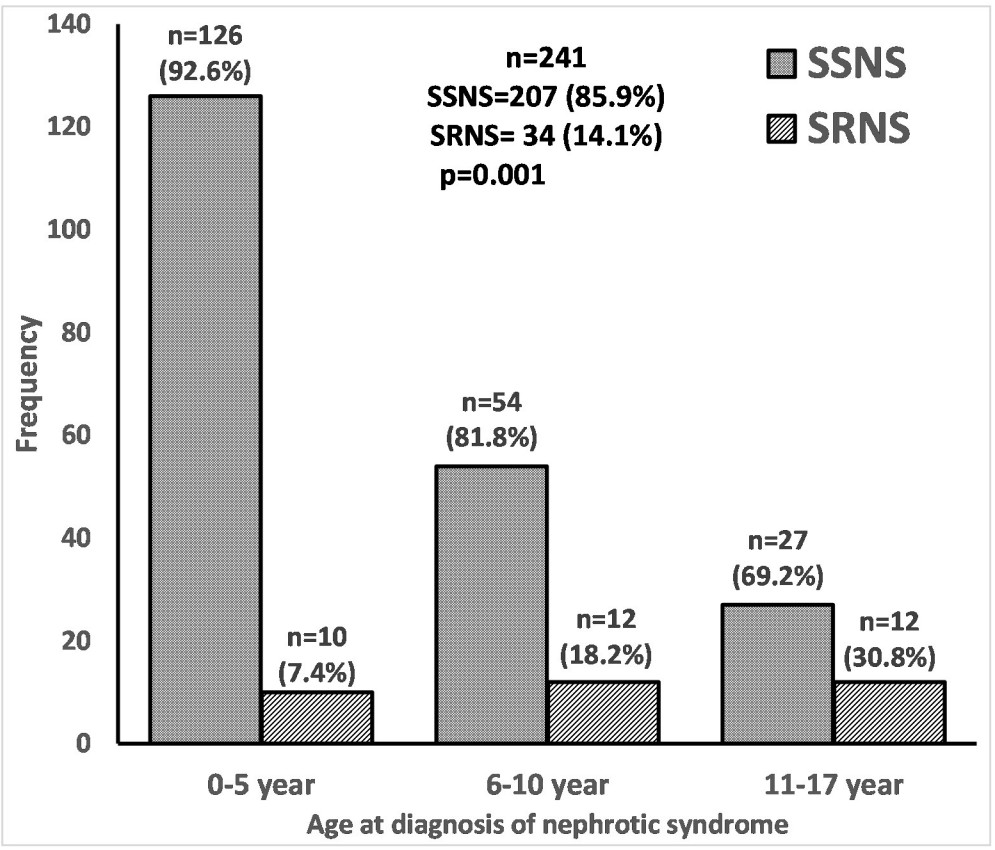

**Fig 3. Distribution of steroid sensitive and resistant nephrotic syndrome with age.**

approximately 70% of White children [9]. In New Orleans, USA, Kim showed that SRNS was four times more common in African Americans compared with children of European ancestry [12].

The reasons for the high steroid sensitivity in our report are unknown but we note that the children in our report were similar in many respects with Asian, European and American children with nephrotic syndrome who historically have high rates of SSNS. Notable among these features is the peak age at onset of nephrotic syndrome in preschool children. Younger age of onset of nephrotic syndrome supports a histologic diagnosis of minimal change disease which is characterised by about 90% steroid sensitivity rate [26]. In our report, children with SSNS were about 4 years younger than those with SRNS at onset of nephrotic syndrome. The diminishing prevalence of infection-related causes of nephrotic syndrome because of better standard of living is another reason that has been proposed. In a review of childhood nephrotic syndrome in tropical Africa, Olowu et al. [14] showed that SSNS as a proportion of all cases of nephrotic syndrome increased from 26.5% in the period between 1960 and 1989 to 66.1% in the era after 1989 which mirrors the diminishing presence of infectious disease causes of nephrotic syndrome. The adoption of 8-week of steroid therapy in 2013 before making a diagnosis of SRNS may have also contributed to the high rate in the present study. We showed higher rate of SSNS in children developing nephrotic syndrome from 2013 than in those managed prior to this time, although the study by Bakhiet in Johannesburg which used a similar definition reported a much lower rate of SSNS.

**Table 1. Demographic and routine biochemical characteristics of children with steroid sensitive and resistant nephrotic syndrome.**

| Characteristics | All children | Steroid responsiveness status | | P value |
| --- | --- | --- | --- | --- |
| | | Sensitive | Resistant | |
| | n = 241 | n = 207 (85.9%) | n = 34 (14.1%) | |
| **Demographics** | | | | |
| Age at diagnosis (IQR), years | 5.1 (3.0–8.7) | 4.8 (2.9–7.9) | 9.3 (4.5–13.8) | <0.001 |
| Gender | | | | 0.353 |
| Female | 96 (39.8) | 80 (83.3) | 16 (16.7) | |
| Male | 145 (60.2) | 127 (87.6) | 18 (12.4) | |
| Tribe | | | | 0.010 |
| Yoruba | 117 (48.5) | 108 (92.3) | 9 (7.7) | |
| Ibo | 81 (33.6) | 65 (80.2) | 16 (19.8) | |
| Hausa | 8 (3.3) | 8 (100.0) | 0 (0.0) | |
| Others | 35 (12.6) | 26 (74.3) | 9 (25.7) | |
| Year of diagnosis of NS | | | | 0.013 |
| 2010–2012 | 42 (17.4) | 31 (73.8) | 11 (26.2) | |
| 2013–2019 | 199 (82.6) | 176 (88.4) | 23 (11.6) | |
| Hospital location | | | | 0.082 |
| LUTH | 176 (73.0) | 147 (83.5) | 29 (16.5) | |
| LASUTH | 65 (27.0) | 60 (92.3) | 5 (7.7) | |
| Duration of follow up, months | 23.0 (8.0–43.5) | 21.4 (6.9–41.6) | 26.0 (14.6–64.0) | 0.036 |
| **At the time of diagnosis** | | | | |
| Elevated SBP or DBP | 68 (34.0) | 57 (83.8) | 11 (16.2) | 0.850 |
| Haematuria, n (%) | 68 (36.7) | 58 (85.3) | 10 (14.7) | 0.714 |
| Serum albumin, mg/dl | 1.40 (1.10–1.86) | 1.37 (1.00–1.86) | 1.50 (1.29–1.88) | 0.106 |
| Serum cholesterol, mg/dl | 420 (319–537) | 416 (319–524) | 436 (331–592) | 0.506 |
| Serum creatinine, mg/dl | 0.50 (0.30–0.80) | 0.44 (0.28–0.71) | 0.70 (0.52–1.02) | <0.001 |
| eGFR, ml/min/1.73 m$^2$ | 93.9 (63.4–165.5) | 101.0 (64.4–175.5) | 74.3 (53.5–94.5) | 0.008 |
| eGFR>140 ml/min/1.73 m$^2$, n (%) | 58 (27.9) | 57 (98.3) | 1 (1.7) | 0.001 |

All continuous variables are medians (interquartile ranges); eGFR: estimated glomerular filtration rate; LUTH: Lagos University Teaching Hospital; LASUTH: Lagos State University Teaching Hospital.

We observed that children belonging to the Ibo and other ethnic groups in Nigeria had lower rate of SSNS compared with those of the Yoruba and Hausa ethnic groups. The reasons for the ethnic differences in steroid response are not known, however, other studies from regions of Nigeria where the Ibos are predominant reported substantially lower steroid sensitivity [27, 28]. In contrast to our finding, a recent study in another southwest region of Nigeria, predominantly populated by the Yorubas, reported that only 60% of the children had SSNS [19]. These observations support a role for both genetics and environmental factors in the pathogenesis of childhood nephrotic syndrome in Nigeria.

Among the routine laboratory parameter, only serum creatinine discriminated children with SSNS from those with SRNS. On the average, children with SSNS had serum creatinine that were 35% lower than those with SRNS. Similarly, the estimated glomerular filtration rate was higher in those with SSNS. While the lower serum creatinine in those with SSNS may be explained by the predominance of SSNS in younger children, the higher estimated GFR values suggest that the kidney function of children with SSNS may be better than those with SRNS. Other studies have observed that children with SRNS have poorer kidney function and more rates of acute kidney injury at diagnosis of nephrotic syndrome than those with SSNS [29, 30].

This is not unexpected because SRNS is more likely associated with a more severe kidney pathology such as focal segmental glomerulosclerosis and membranoproliferative glomerulonephritis than SSNS. On the other hand, similar to other studies, we found no difference in the level of serum albumin and cholesterol at the time of diagnosis in children with SSNS and SRNS.

## Limitations

Using 8 weeks of steroid therapy to define steroid resistance may have inflated the rate of SSNS in the present report. Different definitions for SRSN exist in different regions of the world, but we chose the KDIGO glomerulonephritis guidelines as per our centres' protocol. Although our study does not include all children with nephrotic syndrome in Lagos, being the only two referral hospitals for paediatric kidney diseases in Lagos, we envisage that true rate of SSNS may be higher than we have reported because children with difficult-to-treat nephrotic syndrome are more likely to be referred to our centres and included in this study.

## Conclusion

Our report indicates that nephrotic syndrome in children in Lagos closely resembles the pattern described in children of Asian and European descent with peak in preschool children and high steroid response rate. Even in the same country, response to steroids may vary among different ethnic groups. An obvious implication of this study is that Black children presenting with INS should be managed similar to children of other races.

## Supporting information

**S1 File. Dataset.**
(XLSX)

## Acknowledgments

We are grateful to the resident doctors who helped with the data extraction.

## Author Contributions

**Conceptualization:** Christopher I. Esezobor, Rasheed Gbadegesin.

**Data curation:** Christopher I. Esezobor, Adaobi U. Solarin.

**Formal analysis:** Christopher I. Esezobor.

**Funding acquisition:** Rasheed Gbadegesin.

**Methodology:** Christopher I. Esezobor, Adaobi U. Solarin.

**Supervision:** Rasheed Gbadegesin.

**Writing – original draft:** Christopher I. Esezobor.

**Writing – review & editing:** Christopher I. Esezobor, Adaobi U. Solarin, Rasheed Gbadegesin.

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
