## [Decision Letter · Decision Letter 0]

31 Jul 2020

PONE-D-20-18940

Changing epidemiology of nephrotic syndrome in Nigerian children: A cross-sectional study

PLOS ONE

Dear Dr. Esezobor,

Thank you for submitting your manuscript to PLOS ONE. After careful consideration, we feel that it has merit but does not fully meet PLOS ONE’s publication criteria as it currently stands. Therefore, we invite you to submit a revised version of the manuscript that addresses the points raised during the review process.

We look forward to receiving your revised manuscript.

Kind regards,

Rajendra Bhimma, PhD

Academic Editor

PLOS ONE

Journal Requirements:

2. For studies involving humans categorized by race/ethnicity, age, disease/disabilities, religion, sex/gender, sexual orientation, or other socially constructed groupings, authors should:

1) Explicitly describe their methods of categorizing human populations,

2) Define categories in as much detail as the study protocol allows,

3) Justify their choices of definitions and categories,

4) Explain whether (and if so, how) they controlled for confounding variables such as socioeconomic status, nutrition, environmental exposures, or similar factors in their analysis, and

5) Update outmoded terms and potentially stigmatizing labels to more current, acceptable terminology.

Examples: “Caucasian” should be changed to “white” or “of [Western] European descent” (as appropriate); “XXX victims” should be changed to “patients with XXX.”

3. In your ethics statement in the Methods section and in the online submission form, please provide additional information about the data used in your study. Specifically, please ensure that you have discussed whether all data were fully anonymized before you accessed them and/or whether the IRB or ethics committee waived the requirement for informed consent.

For the children also participating in the genetic study, please clarify whether written informed consent was obtained from parents or guardians of the children.

4. In figures 2 and 3, please indicate your measures of significance either with symbols or with your actual p-values.

Additional Editor Comments (if provided):

The reviewers have raised concerns that need to be addressed before the article can be considered for publication.

Reviewers' comments:

Reviewer's Responses to Questions

**Comments to the Author**

1. Is the manuscript technically sound, and do the data support the conclusions?

Reviewer #1: Partly

Reviewer #2: Partly

2. Has the statistical analysis been performed appropriately and rigorously? 

Reviewer #1: Yes

Reviewer #2: I Don't Know

3. Have the authors made all data underlying the findings in their manuscript fully available?

Reviewer #1: Yes

Reviewer #2: Yes

4. Is the manuscript presented in an intelligible fashion and written in standard English?

Reviewer #1: Yes

Reviewer #2: Yes

5. Review Comments to the Author

Reviewer #1: My reason for suggesting a minor revision rests mainly in how the authors calculated the prevalence of the AKI.You have partly answered this under "Limitations" - see lines 242 to 246 of manuscript. What I find disappointing, is that urine output, urine microscopy, FeNa nor urinary Beta2microglobulin / creatinine ratios were not looked at to substantiate possible AKI .The back calculated serum creatinine would tend to increase the prevalence of AKI, particularly with the the GFR set at 120 ml/min/1.73m2 ie the basal serum creatinine would then be artificially lower, and therefore a 1.5 times increase would more easily be reached, giving an easier overestimate of AKI. Therefore, your estimate of 40.9% of children with AKI at the time of diagnosis, is likely to be an over-estimate. However, AKI affecting the tubules only, could be missed if one does not address the examination of the urine as given above.

I do realize that AKI has been documented at even higher levels, but my concerns above still apply. Please see references 23, 25 and 31 of your paper- AKI often aggravated in the ICU or with the use of nephrotoxic drugs.

On the other hand, over the last seven years, steroid sensitive nephrotic syndrome in Black children in Lagos, particularly in the Yoruba ethnic group, is comparable to the rate in Asians and Caucasians elsewhere.I was surprised that over 80% of the steroid sensitive group were hypertensive. Was this particularly in the younger age group? Were the children quiet when blood pressure done? - adequate cuffs? - upper arm at heart level if child sitting? Was it aggravated by the steroid dosage?

Reviewer #2: The questions regarding the paper are as follows:

1. Are all children int these areas access the teaching hospitals or there are private practises handling these children?

2. Was the cyclophosphamide given IVI or orally?

3. Were the levels of cyclosporin monitored as higher doses cause renal dysfunction due to nephrotoxicity?

4. The back calculation of creatinine is not tested scientifically, the authors quoted for this method reference 25," However, this method of estimating baseline kidney function has not been validated, nor has it been systematically compared with other methods. Substantial interstudy heterogeneity also exists in estimating baseline SCr"

5. Thus my recommendation is to leave out the Acute renal failure objective. the rest of the conclusions are valid according to their study.

6. PLOS authors have the option to publish the peer review history of their article (what does this mean?). If published, this will include your full peer review and any attached files.

Reviewer #1: **Yes: **Peter D Thomson

Reviewer #2: No

---

## [Author Response · Author response to Decision Letter 0]

17 Aug 2020

Dear Editor,

Thank you for giving us the opportunity to have our manuscript reviewed for possible publication in your Journal. Again, we are grateful for the useful feedback that have enriched the message of the manuscript.

We have carefully read all the comments provided and have revised the manuscript accordingly. Below is a point-by-point response to the comments raised during the peer review:

Editor: Please ensure that your manuscript meets PLOS ONE's style requirements, including those for file naming. The PLOS ONE style templates can be found at

Response: We have substantially followed the Journal’s style requirements in the revised manuscript. Specifically, we have revised the font sizes for the various heading levels, removed funding source from acknowledge section, inserted the table and figures immediately after the paragraph where they are first mentioned and renamed the supplementary file.

Editor: For studies involving humans categorized by race/ethnicity, age, disease/disabilities, religion, sex/gender, sexual orientation, or other socially constructed groupings, authors should:

1) Explicitly describe their methods of categorizing human populations,

2) Define categories in as much detail as the study protocol allows,

3) Justify their choices of definitions and categories,

4) Explain whether (and if so, how) they controlled for confounding variables such as socioeconomic status, nutrition, environmental exposures, or similar factors in their analysis, and

5) Update outmoded terms and potentially stigmatizing labels to more current, acceptable terminology.

Examples: “Caucasian” should be changed to “white” or “of [Western] European descent” (as appropriate); “XXX victims” should be changed to “patients with XXX.”

Response: “Caucasian” has been replaced everywhere it appeared in the manuscript with “children of European descent” or “children with European ancestry”

Editor: In your ethics statement in the Methods section and in the online submission form, please provide additional information about the data used in your study. Specifically, please ensure that you have discussed whether all data were fully anonymized before you accessed them and/or whether the IRB or ethics committee waived the requirement for informed consent.

For the children also participating in the genetic study, please clarify whether written informed consent was obtained from parents or guardians of the children.

Response: data for about 67% of the children included in this report were collected as part of their participation in a genetic study on childhood nephrotic syndrome (genetic analysis is still ongoing). For these children, their caregivers provided written informed consent; children older than 12 years also provided assent. For the other 1/3rd of the participants included in this study, the requirement for consent was waived because (1.) only information recorded in their medical records as part of routine patient care was assessed, and (2.) the vast majority of these patients were no longer in active follow up either because the family has moved out of town, nephrotic syndrome is cured or child had died. Furthermore, all retrieved medical information was anonymised to maintain confidentiality. The ethic statement in the manuscript has been improved upon to provide this clarity

Editor: In figures 2 and 3, please indicate your measures of significance either with symbols or with your actual p-values.

Response: The p values have been added to the two figures

Additional Editor Comments (if provided):

The reviewers have raised concerns that need to be addressed before the article can be considered for publication.

Reviewers' comments:

Reviewer's Responses to Questions

Comments to the Author

1. Is the manuscript technically sound, and do the data support the conclusions?

Reviewer #1: Partly

Reviewer #2: Partly

2. Has the statistical analysis been performed appropriately and rigorously?

Reviewer #1: Yes

Reviewer #2: I Don't Know

3. Have the authors made all data underlying the findings in their manuscript fully available?

Reviewer #1: Yes

Reviewer #2: Yes

4. Is the manuscript presented in an intelligible fashion and written in standard English?

Reviewer #1: Yes

Reviewer #2: Yes

5. Review Comments to the Author

Reviewer #1: My reason for suggesting a minor revision rests mainly in how the authors calculated the prevalence of the AKI.You have partly answered this under "Limitations" - see lines 242 to 246 of manuscript. What I find disappointing, is that urine output, urine microscopy, FeNa nor urinary Beta2microglobulin / creatinine ratios were not looked at to substantiate possible AKI .The back calculated serum creatinine would tend to increase the prevalence of AKI, particularly with the the GFR set at 120 ml/min/1.73m2 ie the basal serum creatinine would then be artificially lower, and therefore a 1.5 times increase would more easily be reached, giving an easier overestimate of AKI. Therefore, your estimate of 40.9% of children with AKI at the time of diagnosis, is likely to be an over-estimate. However, AKI affecting the tubules only, could be missed if one does not address the examination of the urine as given above.

I do realize that AKI has been documented at even higher levels, but my concerns above still apply. Please see references 23, 25 and 31 of your paper- AKI often aggravated in the ICU or with the use of nephrotoxic drugs.

Response: We agree with the comments of both reviewers that using back-calculation to determine serum creatinine may have substantial inherent weakness. As a result, we have removed all aspects of the manuscript detailing the methods, results and discussion of acute kidney injury. However, we still retained aspects of the manuscript that relates to serum creatinine and estimated glomerular filtration rate values

Reviewer #1: On the other hand, over the last seven years, steroid sensitive nephrotic syndrome in Black children in Lagos, particularly in the Yoruba ethnic group, is comparable to the rate in Asians and Caucasians elsewhere.I was surprised that over 80% of the steroid sensitive group were hypertensive. Was this particularly in the younger age group? Were the children quiet when blood pressure done? - adequate cuffs? - upper arm at heart level if child sitting? Was it aggravated by the steroid dosage?

Response: The correct reading of the results on Table 1 is: 27.5% (57) of the of 207 children with SSNS had elevated BP compared with 32.4% (11) of the 34 children with SRNS. This is so when the results are read down the columns rather than across the rows.

Reviewer #2: The questions regarding the paper are as follows:

1. Are all children int these areas access the teaching hospitals or there are private practises handling these children?

Response: there are several private hospitals in Lagos and they may have some children with nephrotic syndrome. However, two of the authors of these manuscripts (CIE and AUS) are the only active paediatric nephrologists in the city. Our experience is that children with difficult-to-treat nephrotic syndrome are more likely to be referred to our centres rather than managed in the other hospitals without a paediatric nephrologist

Reviewer #2. Was the cyclophosphamide given IVI or orally?

Response: Initially cyclophosphamide was being given as monthly intravenous dose but in the latter years that the study period covered, it was mostly administered orally

Reviewer #2. Were the levels of cyclosporin monitored as higher doses cause renal dysfunction due to nephrotoxicity?

Response: Yes, cyclosporine drug levels were monitored, but not more frequently than every 3 months after the first month of starting. The frequency of monitoring depended on ability of the family of the child to afford it

Reviewer #2. The back calculation of creatinine is not tested scientifically, the authors quoted for this method reference 25," However, this method of estimating baseline kidney function has not been validated, nor has it been systematically compared with other methods. Substantial interstudy heterogeneity also exists in estimating baseline SCr"

5. Thus my recommendation is to leave out the Acute renal failure objective. the rest of the conclusions are valid according to their study.

Response: We agree with this recommendation and have removed the aspect of AKI prevalence from all aspects of the manuscript.

---

## [Decision Letter · Decision Letter 1]

3 Sep 2020

Changing epidemiology of nephrotic syndrome in Nigerian children: A cross-sectional study

PONE-D-20-18940R1

Dear Dr. Christopher Imokhuede Esezobor

We’re pleased to inform you that your manuscript has been judged scientifically suitable for publication and will be formally accepted for publication once it meets all outstanding technical requirements.

An invoice for payment will follow shortly after the formal acceptance. To ensure an efficient process, please log into Editorial Manager at http://www.editorialmanager.com/pone/, click the 'Update My Information' link at the top of the page, and double-check that your user information is up-to-date. If you have any billing-related questions, please contact our Author Billing department directly at authorbilling@plos.org.

Kind regards,

Rajendra Bhimma, PhD

Academic Editor

PLOS ONE

Additional Editor Comments (optional):

Reviewers' comments:

Reviewer's Responses to Questions

**Comments to the Author**

1. If the authors have adequately addressed your comments raised in a previous round of review and you feel that this manuscript is now acceptable for publication, you may indicate that here to bypass the “Comments to the Author” section, enter your conflict of interest statement in the “Confidential to Editor” section, and submit your "Accept" recommendation.

Reviewer #1: All comments have been addressed

Reviewer #2: All comments have been addressed

2. Is the manuscript technically sound, and do the data support the conclusions?

Reviewer #1: Yes

Reviewer #2: Yes

3. Has the statistical analysis been performed appropriately and rigorously? 

Reviewer #1: I Don't Know

Reviewer #2: N/A

4. Have the authors made all data underlying the findings in their manuscript fully available?

Reviewer #1: Yes

Reviewer #2: Yes

5. Is the manuscript presented in an intelligible fashion and written in standard English?

Reviewer #1: Yes

Reviewer #2: Yes

6. Review Comments to the Author

Reviewer #1: Line 167:

74.3 ml/min/1.73* ) and more hyperfiltration( 98.3% vs 1.7% ) than those with SRNS.

* need to put in " m2 "ie metres squared. Also, the percentages 98.3 and 1.7, taken from the Table: Demographics and routine biochemical characteristics with steroid-sensitive and resistant nephrotic syndrome ( this heading of the Table on lines 170 and 171), refers to the 58 patients with hyperfiltration = 57/207( 27.5% SSNS ), and 1/34( 2.94% SRNS ) ie the figs given in the text and table gives an exaggerated picture, whereas 27.5 % gives the actual percentage of patients with SSNS who seemed to have hyperfiltration, and likewise 2.94% of patients with SRNS - in their respective columns( SSNS vs SRNS ), and therefore the percentages of those with hyperfiltration should also be given in their respective groups in the Table as well.

Also in the same Table, the Duration of follow up in months, the SSNS group has a median of 21.4 months( range 6.9 - 41.6), and the SRNS group a median of 26.0 months and a range of ( 14.6 - 64.0 ); and comparing the follow-up periods of the two groups statistically, the difference reaches significance: p-value 0.036. Hence, the SSNS group has a shorter follow up period, with a presumptive diagnosis of minimal change nephrotic syndrome, with a longer follow up, some may be less steroid-responsive, and if a biopsy was done on these, the histological diagnosis may be focal glomerulosclerosis, rather than minimal change disease.( see reference below)

However, this does not detract from the authors' finding that SSNS in black( particularly the Yoruba ethnic group) is as prevalent as in Asians and Europeans.

Ref = Renal problems in black South African children, P.D.Thomson. Pediatr Nephrol 1997 Vol 11; 508-512, but SSNS and minimal change disease in black SA children more prevalent recently**, as given by the authors in their ref by Bakhiet et al., where many of the patients were also biopsied.** but not as much as in Nigerian children.

Reviewer #2: All the editorial & reviewer's comments have been adequately answered.

A new question. Were there ant adverse or side effects of steroid therapy? Because from the commencement of the steroid therapy to the end of the course is approximately 4 to 5 months when it is stopped. I am curious to know if they observed any side effects of steroids? But it is not part of the study and i would not have any objections if it is not commented on.

I am happy with the script with all the adjustments.

7. PLOS authors have the option to publish the peer review history of their article (what does this mean?). If published, this will include your full peer review and any attached files.

Reviewer #1: **Yes: **Peter D Thomson

Reviewer #2: No

---

## [Editor Report · Acceptance letter]

10 Sep 2020

PONE-D-20-18940R1

Changing epidemiology of nephrotic syndrome in Nigerian children: A cross-sectional study

Dear Dr. Esezobor:

I'm pleased to inform you that your manuscript has been deemed suitable for publication in PLOS ONE. Congratulations! Your manuscript is now with our production department.

Kind regards,

on behalf of

Professor Rajendra Bhimma 

Academic Editor

PLOS ONE